# Research priorities in children requiring elective surgery for conditions affecting the lower limbs: a James Lind Alliance Priority Setting Partnership

Martinique Vella-Baldacchino,[1] Daniel C Perry,[1] Andreas Roposch,[2] Nicholas Nicolaou,[3] Stephen Cooke,[4] Patricia Ellis,[5] Tim Theologis [6]

[1]Nuffield Department of Orthopaedics Rheumatology and Musculoskeletal Science, Botnar Research Centre, Oxford, UK
[2]University College London Great Ormond Street Institute of Child Health Library, London, UK
[3]Paediatric Orthopaedic Surgery, Sheffield Children's NHS Foundation Trust, Sheffield, UK
[4]Paediatric Orthopaedic Surgery, University Hospital Coventry, Coventry, UK
[5]James Lind Alliance, Southampton, UK
[6]Paediatric Orthopaedic Surgery, Oxford University Hospitals NHS Foundation Trust, Oxford, UK

**Correspondence to**
Tim Theologis;
timtheologis@gmail.com

## ABSTRACT

**Objective** To identify and prioritise research questions concerning the elective surgical management of children with conditions affecting the lower limb by engaging patients, carers and healthcare professionals.

**Design** A modified nominal group technique.

**Setting** UK.

**Participants** 388 individuals (29 patients, 155 parents/carers, 204 healthcare professionals) were recruited through hospital clinics, patient charities and professional organisations and participated in the initial prioritisation survey; 234 individuals took part in the interim prioritisation survey. 33 individuals (3 patients, 9 parents/carers, 11 healthcare professionals, 7 individuals representing the project's steering group and 3 James Lind Alliance (JLA) facilitators) attended the final face-to-face workshop to rank the top 10 research priorities.

**Interventions** Surveys were distributed using various media resources such as newsletters, internet messaging boards and the 'Paediatric Lower Limb Surgery Priority Setting Partnership (PSP) website. Printed copies of the questionnaire were also made available to families in outpatient clinics.

**Outcome measures** Survey results, top 10 and top 26 priority rankings

**Results** The process took 18 months to complete (July 2017–January 2019); 388 people generated 1023 questions; a total of 801 research questions were classified as true uncertainties. Following the JLA methodology, 75 uncertainties were developed from the initial 801 questions. Twenty six of those were selected through a second survey and were taken to the final face-to-face workshop where the top 10 research priorities were selected. The top10 priorities included questions on cerebral palsy, common hip conditions (ie, Perthes' disease and developmental dysplasia of the hip) as well as rehabilitation techniques and methods to improve shared decision-making between clinicians and patients/families.

**Conclusions** This is the first JLA PSP in children's orthopaedic surgery, a particularly under-researched and underfunded area. We have identified important research topics which will guide researchers and funders and direct their efforts in future research.

### Strengths and limitations of this study

► This was the first national research prioritisation exercise to identify research priorities among children affected by lower limb conditions.
► There was balanced representation of patients, parents/carers and healthcare professionals.
► There was a broad range of questions representing a variety of conditions and treatments.
► Important questions applicable to the entire range of paediatric lower limb conditions were identified.
► As a result of the wide variety of conditions included in this Priority Setting Partnership, rare conditions did not make it to the top 10 research priority list despite the evident research gaps.

## INTRODUCTION

Musculoskeletal symptoms are the primary reason for referral to paediatric outpatient clinics. Each year, one in eight children visits the doctor for musculoskeletal conditions, some of which are responsible for long-term impairment and disability.[1] While musculoskeletal symptoms are common, the evidence underpinning their management is poor. In particular, orthopaedic surgical practice in children is almost exclusively based on poor-quality evidence.[2–5] This poor-quality evidence has led to significant variation in surgical practices nationally and internationally.[2–5] This variation has resulted in conflicting information and loss of confidence in treatment pathways and, sometimes, in the clinicians who deliver them.

Clinical research may not truly represent the perceptions of clinicians' or patients' about which research questions are most important.[6] In 2003, Ian Chalmers published an idea in the Lancet promoting engagement, communication and discussion between researchers, patients, the public, carers and clinicians to agree on which treatment

uncertainties mattered most to them and thus set formal future research priorities.[6] This became known as the James Lind Alliance (JLA), which has since then evolved and spread.[7] This has given the opportunity to patients and members of the public to have an equal voice as clinicians and researchers in influencing the research agenda.

The JLA's infrastructure is funded by the National Institute for Health Research (NIHR) and which oversees the overall process in a transparent and structured manner.[8] NIHR encourages active involvement of the public and supports JLA in feeding back research priorities to national funding bodies. There have been more than 60 JLA projects investigating a range of treatment uncertainties, working with patients, carers and healthcare professionals and focusing on various clinical topics such as scoliosis, spinal cord injury, surgery for common shoulder problems and joint replacements for osteoarthritis.[9]

Lower limb pathology forms the bulk of the elective practice in children's orthopaedic diseases, and these diseases share similar functional limitations in mobility. The aim of the Paediatric Lower Limb Surgery (PLLS) Priority Setting Partnership (PSP) was to identify the unanswered questions on elective surgery for lower limb conditions in the paediatric population and agree by consensus on research priorities by forming a partnership between patients, parents/carers and clinicians.

## METHOD
We followed the modified nominal group technique outlined in the step-by-step guidelines for the conduct of a JLA PSP. This technique has tested methods for PSPs to work effectively and reach credible and useful outcomes. In order for JLA to fully endorse the final top10 research priorities, the founding principles of patient and clinician involvement, transparency and systematic rigour had to be respected.[10] A JLA adviser (PE) was appointed by the NIHR Evaluation, Trials and Studies Coordinating Centre to support and guide the PSP setting process and liaise with the clinical lead. A steering group, consisting of charity representatives (LGW, EM), parents (HGO,CAG), patients (DD), physiotherapists (CD, CB), allied health professionals (CW, EW), paediatric–orthopaedic surgeons (DCP, AR, NN, SC), a surgical trainee (MV-B), the JLA advisor (PE) and the JLA PSP administrator (CR) was appointed by the PSP lead. JLA PLLS PSP meetings were organised between July 2017–January 2019.

### Patient and public involvement
This project was supported closely by patients, parents and organisations that represented them. Indeed, these organisations drove much of the study as they recognise the huge variability in orthopaedic practice and identify that this makes it difficult to offer advice as each surgeon seemingly treats the same disease differently. Patient representatives were appointed to the steering group and contributed to the design and the smooth running of the study. As with all JLA PSPs, involvement of patients, parents and carers continued throughout the project, facilitated by the team

and the JLA advisor. Approximately half of the submitted research questions were from the patients and the public. Dissemination was delivered by both professional and lay members of the steering group through a variety of media such as infographics, a project report, conference presentations and online social media.

### Ethics
Consideration was given to applying for ethical approval for this study. Most PSPs do not require ethical approval as no personally identifiable data are stored (http://www.jla.nihr.ac.uk/jla-guidebook/chapter-5/consent-and-ethics.htm). We tested our study against the Health Research Authority criteria and were advised to proceed without an application for ethics approval.

### Partner organisations: identification and invitation
Partner organisations were identified through a process of peer knowledge and through the networks of the steering group members. The organisations were invited to participate via a communication package describing the JLA PLLS PSP objectives and process. The partners were first asked to provide their views and feedback on the proposed protocol of the partnership.

Organisations represented paediatric–orthopaedic patients and their families or carers as well as relevant healthcare professionals, including medical doctors, nurses, physiotherapists and other allied health professionals with clinical experience of paediatric surgery for conditions affecting the lower limbs. Children and young people under the age of 16 years were encouraged to voice their views separately from those of their parents. Parents were asked to encourage their children to fill in separate survey forms.

### Identifying treatment uncertainties
An online survey, agreed by the steering group, was set up and the link distributed to partner organisations. These organisations were encouraged to freely distribute the survey link and solicit research questions and uncertainties from their members. The steering group further encouraged the submission of questions from a broad array of individuals from across the society using a variety of media, including newsletters, internet message boards and postal questionnaires. The link for the survey was available through the PLLS PSP website. Where printed copies of the questionnaire were made available, the data was entered into the online survey. The steering group monitored the responses to the survey and underrepresented groups were targeted while the survey was live (January 2018–March 2018).

### Refining questions and uncertainties
The raw questions collected during the initial survey were organised into broad lower limb condition categories. Questions which did not relate to the JLA PLLS objectives were excluded and labelled as 'Out of Scope'. The in-scope raw questions were further grouped under general indicative questions. The in-scope questions were then searched

using evidence published by National Institute for Health and Care Excellence, Cochrane library, systematic reviews and randomised control trials (levels I and II). Only evidence published in the English language over the past 10 years was inlcuded. A question was confirmed as a genuine uncertainty if it could not be answered using the literature-search method described above. Several topic experts in various fields were consulted to finalise the decisions. The full list of questions submitted through the initial survey can be found on the web (http://www.jla.nihr.ac.uk/priority-setting-partnerships/Paediatric-lower-limb-surgery/downloads/Paediactric-Lower-Limb-Surgery-PSP-final-data-sheet.pdf).

In-scope questions, which could be resolved with reference to existing literature, the 'unrecognised knowns', were identified and listed. These were then passed on to relevant partners within the steering group in order to communicate the information to the appropriate patient groups. By merging and grouping the remaining true uncertainties on similar themes together, the final number of research uncertainties was reduced to 75. These 75 questions were agreed to by the steering group through a consensus process and were entered into the next stage of prioritisation. A list of the 75 questions can be found online (http://www.jla.nihr.ac.uk/priority-setting-partnerships/Paediatric-lower-limb-surgery/downloads/Paediactric-Lower-Limb-Surgery-PSP-final-data-sheet.pdf).

## Prioritisation—interim stage

The long list of 75 questions was reduced to a shorter list by a further online survey of the same partners, whereby respondents (a mix of patients, parents and healthcare professionals) were asked to identify the 10 most important questions. Participants were asked to submit their preferences between 22 August 2018 and 29 September 2018. The steering group reviewed the results of the second survey and agreed on the top 26 questions to be taken to the final prioritisation meeting. This followed the JLA recommendation to select approximately 25 questions for the final prioritisation. Care was taken to adequately represent the top choices of children, parents/carers and professional groups.

## Prioritisation—final prioritisation workshop

The top 26 questions, which were most frequently selected in the interim prioritisation stage, were taken to the final face-to-face workshop. Thirty individuals, representing all relevant partners, were invited and asked to declare any conflicts of interests a priori. Care was taken to respect participants' views and ensure confidentiality of each individual's responses.

Discussions were led by three independent JLA facilitators who had no previous experience in PLLS. The established JLA process was followed to reach consensus.[10]

## RESULTS

Figure 1 outlines the stages and the process of the JLA PLLS PSP. The initial survey generated a total of 1023

Survey (January 2018 – March 2018)
388 individuals submitted 1023 questions
47% individuals were patients, parents or carers

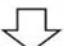

Organising and identifying uncertainties (March 2018 – July 2018)

801 questions classified as in-scope
222 questions were out of scope

75 indicative research questions were generated, all of which were confirmed uncertainties following merging, grouping and literature review.

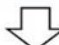

Interim prioritisation online survey (August 2018 – September 2018)

234 individuals selected their top 10 questions from the 75 indicative questions

Final Prioritisation face-to-face workshop (November 2018)

Attended by 30 individuals (6 physiotherapists, 9 parents, 4 patients, 7 orthopaedic surgeons, 1 clinical scientist, 1 advanced nurse practitioner and 2 charity representatives)

Participants asked to rank the top 26 questions from the interim prioritisation survey.

**Figure 1** Summary of the James Lind Alliance Paediatric Lower Limb Surgery Priority Setting Partnership.

questions from 388 individuals consisting of a mix of patients, carers, nurse educators, advanced nurse practitioners, clinical or healthcare scientists and clinicians (paediatricians, physicians, physiotherapists, occupational therapists, nurses, general practitioners and orthopaedic surgeons). Patients, parents and carers represented 47% of the respondents.

From the initial questions, 222 were marked as out of scope, as they did not adhere to the objectives of the JLA PLLS PSP (eg, questions relating to spine or upper limb conditions). A total of 801 uncertainties were classified as in-scope by the steering group. The in-scope uncertainties were grouped into a preliminary list of 75 broader research questions. These were all confirmed as true uncertainties following a literature search, based on the JLA criteria and through consultation with topic experts.

The 75 research questions crystalised through this process were then prioritised through the interim survey involving parents/carers, patients and healthcare professionals. A total of 234 individuals participated in the interim survey and selected 10 questions each which they valued as the most important uncertainties. Of these participants, 117 (50%) were patients or parents, while the remaining were healthcare professionals.

Based on the interim prioritisation the steering group produced a shortlist of 26 uncertainties that were taken forward to the final face-to–face prioritisation workshop held in Oxford on 17 November 2018. A total of

**Box 1  The top 10 research questions**

1. What are the best ways to measure outcome following lower limb orthopaedic surgery in children?
2. What should children's rehabilitation following orthopaedic surgery to the lower limbs include, how long is it expected to last and how does it affect the result of treatment?
3. What is the short-term and long-term clinical and cost effectiveness of orthopaedic lower limb surgery for children with Cerebral Palsy who can walk (considering best timing and technique)?
4. What are the short term and long term outcomes of surgery compared to non-surgical care in the treatment of Perthes disease?
5. What is the role of pre-operative rehabilitation in children presenting with lower limb orthopaedic conditions?
6. What is the short-term and long-term clinical and cost effectiveness of Selective Dorsal Rhizotomy (SDR) in children with Cerebral Palsy who can walk?
7. Can surveillance and non-surgical treatment (physiotherapy, botulinum toxin injections, functional electrical stimulation, orthotics, casting) prevent the development of deformity requiring surgery in children with Cerebral Palsy?
8. What is the best method of screening for Developmental Dysplasia of the Hip (DDH) in terms of clinical and cost effectiveness?
9. What are the best strategies to optimise communication of information between patients/carers and clinicians in order to enable shared decision making ?
10. What is the best management for hip displacement in children with Cerebral Palsy?

30 individuals consisting of 6 physiotherapists, 9 parents, 4 patients, 7 orthopaedic surgeons, 1 clinical scientist, 1 advanced nurse practitioner and t2 charity representatives attended this workshop. Discussions were facilitated by three independent JLA facilitators and the top 10 unanswered research questions were agreed to by all stakeholders present (box 1).

### Publicity

On 25 January 2019, the steering group agreed on a dissemination strategy, using a variety of media such as an infographic, project report, conference presentations, online social media and the publication of a scientific paper.

### DISCUSSION

JLA developed a transparent method to allow patients, carers and healthcare professionals to come together and establish a research agenda. Questions posed by patients or carers are given the same weighted importance to those submitted by healthcare professionals or academic scientists. Patient and public involvement is now recognised as a best practice and is an essential requirement for research funding allocation by funders in the UK, Australia and the USA.[11]

This JLA process has identified the top 10 unanswered research priorities for surgery in children with conditions affecting the lower limbs. The questions identified through this JLA PSP require high quality research that will adequately address these uncertainties. The notable engagement of professionals, patients and the public, will ensure that the questions have a broad reach in terms of real-world impact.

The number one priority was to identify the best way to measure outcomes following lower limb paediatric–orthopaedic surgery. This highlights the importance of developing robust tools to be used in research to process and make informed decisions about clinical effectiveness.

Four of the top 10 priorities were directly related to the management of children with cerebral palsy. This is not surprising, as children with cerebral palsy often undergo interventions for the lower limbs and form a large part of paediatric–orthopaedic surgical practice. The common hip conditions of childhood (Perthes' disease and developmental dysplasia of the hip), as well as rehabilitation techniques and methods to improve shared decision-making between clinicians and patients/families, also contributed to the top 10 list.

The priorities identified through this project are different from those identified through a Delphi process involving paediatric–orthopaedic surgeons.[12] While the surgeons' questions focused around the management of specific conditions, the JLA PSP top priorities also included generic questions on outcome measures, rehabilitation and access/communication. These differences highlight the importance of involving all relevant stakeholders, including those affected by the condition(s), when considering research priorities.

For those considering organising a JLA PSP, data organisation and management is a pivotal determinant to ensure the smooth running and success of the PSP. Adequate data management allows the linking of the questions that are developed during the later stages of the process with the originally submitted uncertainties and the stakeholders groups who submitted them. This is a JLA requirement which underpins clarity and transparency.

The PSP received over 1000 questions with balanced representation from patients and parents. In both the initial and the interim prioritisation surveys, approximately 50% of submissions were from patients or parents, indicating that they engaged well with our JLA PSP. We are confident that the final top 10 research priorities are representative of broad stakeholder involvement from patients, professionals and the wider public.

Participants were chosen to widely represent parent/patient groups and stakeholders, and could only choose from a preprioritised list of 26 questions. They were required to submit an expression of interest in taking part and a declaration of any vested interest of opinion. Furthermore, the final prioritisation meeting was supervised by three independent JLA advisers. The JLA advisers' role is to support and guide the meeting as expert neutral facilitators, ensuring that the process is followed in a fair, transparent way, with equal input from patients, carers and clinicians and their representatives, and equal consideration and

debate are given to each question prioritised. The key role of the JLA adviser is to encourage the understanding of the differing perspectives of patients, carers and clinicians and to ensure that all voices are heard.

High quality research in PLLS is a challenge, as many of the fundamental elements required to deliver a high-quality clinical trial are unknown; that is, disease frequency, agreed outcomes, surgeon equipoise and surgical variation.[2 3] However, the IDEAL (Idea, Development, Exploration, Assessment, Long term Follow-Up) collaborative network has outlined a five-stage approach for assessing surgical interventions. IDEAL has encouraged the use of alternative study designs and prospective cohorts when randomised control trials are not feasible.[13] The IDEAL framework will provide the basis from which the evidence synthesis for many of the questions posed by this PSP will commence.

In recent years, there has been a cultural shift among surgeons in the UK towards evidence-based practice and evidence synthesis. We hope that the clear direction offered by this JLA PSP will enable clinicians, funders, researchers and patients to unite and urgently deliver robust answers to the questions highlighted by this PSP. Additional low quality studies will only add to the noise within the literature and, as such, it is now time for definitive research to occur.

**Acknowledgements** The authors are grateful to a large number of people including all the steering group members: Camille Rougelot, Craig Walsh, Christine Douglas, Catherine Barry, Elizabeth Wright, Emma Morley, Loredana Guetg-Wyatt, Bert Martin, Helen Gregory-Osborn, Daniel Dolley and Catherine Ann Greaves. The authors would also like to thank all the patients and carers for contributing their questions and/or taking part in the interim survey and final workshop, all members of professional or charity organisations who contributed questions and invited patients to contribute, National Institute for Health Research Evaluation, Trials and Studies Coordinating Centre for supporting and approving this Priority Setting Partnership (PSP), British Society of Children's Orthopaedic Surgery (BSCOS), the British Orthopaedic Association (BOA) and the Oxford Biomedical Research Unit for funding the PSP. The main partner organisations: BSCOS, BOA, Oxford Biomedical Research Centre and STEPS Charity provided the foundation for this work.

**Contributors** TT, PE, DCP, AR, NN, SC and MV-B each made substantial contributions to this work, including study design, data collection, analysis and interpretation. MV-B drafted the initial manuscript and all authors (TT, PE, DCP, AR, NN and SC) were involved in revising the manuscript and gave final approval for the version to be published. All authors had access to all data in the study and take responsibility for the integrity of the data and the accuracy of the data analysis.

**Funding** This study and process was funded by the British Society of Children's Orthopaedic Surgery, the British Orthopaedic Association and the Oxford Biomedical Research Centre.

**Competing interests** None declared.

**Patient consent for publication** Not required.

**Provenance and peer review** Not commissioned; externally peer reviewed.

**Data availability statement** Data are available in a public, open access repository. The complete dataset is available on the James Lind Alliance website (http://www. jla.nihr.ac.uk/priority-setting-partnerships/Paediatric-lower-limb-surgery/).

**ORCID iD**
Tim Theologis http://orcid.org/0000-0002-4758-9081

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
