## [Reviewer comments · BMJ Open]

ARTICLE DETAILS

TITLE (PROVISIONAL)	Research priorities in children requiring elective surgery for conditions affecting the lower limbs: a James Lind Alliance Priority Setting Partnership
AUTHORS	Vella-Baldacchino, Martinique; Perry, Daniel; Roposch, Andreas; Nicolaou, Nicholas; Cooke, Stephen; Ellis, Patricia; Theologis, Tim

VERSION 1 - REVIEW

REVIEWER	Anthony Cooper University of British Columbia Canada
REVIEW RETURNED	11-Sep-2019

GENERAL COMMENTS	This is a very important paper which sets out to determine the top 10 research questions in children's lower limb surgery. The authors should be commended for performing such an important project. I would appreciate a comment as to why the paper is limited to lower limb surgery and does not include all paediatric orthopaedic surgery- presumably a parallel study is ongoing/planned? There are no comments on limitations of this study. For example how was biased managed in the workshop- there were 30 individuals here- how did the facilitators ensure that the voices of each individual were equally balanced- did the patients and parents prioritize their personal priorities over that of the population as a whole. Presumably the patients were children-this workshop scenario could be daunting for young people to accurately articulate their concerns If the limitations of this type of project could be expanded upon I think this paper should be accepted
--

REVIEWER	Nicole Williams Women's and Children's Hospital, Adelaide, Australia University of Adelaide, Australia
REVIEW RETURNED	28-Oct-2019

GENERAL COMMENTS	Thank you for an excellent, interesting paper that challenges us all to think about how surgical research is conducted. I have only a few minor comments.
---

	Introduction: Page 4 line 14 please change "pubic" to "public" Method: page 4 line 48 requires capital for "Coordinating" Results: I would have liked to see the initial survey as an Appendix or link on the website. Additionally, the list of 75 questions mentioned at Results page 6 line 26 is not immediately obviously identified from the list of documents found by following the website link. Acknowledgements: please rewrite page 10, line 38 "The authors would also like to thank all the participants who took all the patients and carers for contributing their questions and/or taking part in the interim survey and final workshop..." as this currently doesn't read fluently.
--	---

VERSION 1 – AUTHOR RESPONSE

Reviewer 1

Comment 1: I would appreciate a comment as to why the paper is limited to lower limb surgery and does not include all paediatric orthopaedic surgery- presumably a parallel study is ongoing/planned?

Response: The clinical subject of a James Lind Alliance partnership has to be homogeneous enough to allow a meaningful prioritisation and wide enough to attract sufficient interest. There is considerable common ground in paediatric orthopaedic conditions that affect the lower limb(s): deformity, joint stiffness, pain and impaired mobility are the common problems associated with these conditions. Although the aetiology of these conditions varies considerably, the clinical problems and the resulting motor disability are similar and this is what matters from the patient/family perspective. Therefore, the clinical subject of this work was clearly identifiable. It was decided to rule out upper limb and spinal conditions as these represent a different clinical entity with different symptoms and disability. It was also decided to exclude trauma as the treatment pathway and recovery course are different.

Action: the following comment was added to the last paragraph of the Introduction section: "Lower limb pathology forms the bulk of the elective practice in children's orthopaedic diseases, and these diseases share similar functional limitations in mobility."

Comment 2: There are no comments on limitations of this study. For example how was bias managed in the workshop- there were 30 individuals here- how did the facilitators ensure that the voices of each individual were equally balanced- did the patients and parents prioritize their personal priorities over that of the population as a whole. Presumably the patients were children-this workshop scenario could be daunting for young people to accurately articulate their concerns

Response: Participants were chosen to widely represent parent/patient groups and stakeholders, and could only choose from a pre-prioritised list of 26 questions. They were required to submit an expression of interest in taking part and a declaration of any vested interest of opinion. Furthermore,

the final prioritisation meeting was supervised by three independent James Lind Alliance (JLA) Advisers. The JLA Advisers' role is to support and guide the meeting as expert neutral facilitators, ensuring that the process is followed in a fair, transparent way, with equal input from patients, carers and clinicians and their representatives, and equal consideration and debate given to each question prioritised. A key role of the JLA Adviser is to encourage understanding of the differing perspectives of patients, carers and clinicians, and to ensure that all voices are heard.

Action: the above was added to the discussion as a potential limitation.

Reviewer 2

Introduction: Page 4 line 14 please change "pubic" to "public"

Corrected, thank you

Method: page 4 line 48 requires capital for "Coordinating"

Corrected, thank you

Results: I would have liked to see the initial survey as an Appendix or link on the website. Additionally, the list of 75 questions mentioned at Results page 6 line 26 is not immediately obviously identified from the list of documents found by following the website link.

Direct links to the initial survey questions and the 75 questions of the final prioritisation were provided in the text under Results/ Refining Questions and Uncertainties.

Acknowledgements: please rewrite page 10, line 38 "The authors would also like to thank all the participants who took all the patients and carers for contributing their questions and/or taking part in the interim survey and final workshop..." as this currently doesn't read fluently.

Corrected, thank you

VERSION 2 – REVIEW

REVIEWER	Anthony Cooper BC Children's Hospital University of British Columbia Canada
REVIEW RETURNED	12-Nov-2019

GENERAL COMMENTS	All concerns addressed
------------------------